# Addressing Vaccine Hesitancy in College Students Post COVID-19 Pandemic: A Systematic Review Using COVID-19 as a Case Study

**DOI:** 10.3390/vaccines13050461

**Published:** 2025-04-25

**Authors:** Wai Yan Min Htike, Muxuan Zhang, Zixuan Wu, Xinyu Zhou, Siran Lyu, Yiu-Wing Kam

**Affiliations:** Division of Natural and Applied Science, Duke Kunshan University, No. 8 Duke Avenue, Kunshan 215316, China; wy89@duke.edu (W.Y.M.H.); mz229@duke.edu (M.Z.); zwu102@jh.edu (Z.W.); xinyu.zhou@dukekunshan.edu.cn (X.Z.); siran.lyu@dukekunshan.edu.cn (S.L.)

**Keywords:** vaccine hesitancy, COVID-19 vaccine, confidence, convenience, complacency

## Abstract

**Background**: Resistance to vaccinations continues to pose a considerable challenge to attaining widespread vaccination, especially among the college student demographic, who are pivotal in championing public health initiatives. This systematic review investigates the elements that influence reluctance to receive the COVID-19 vaccine among university students globally. Utilizing the WHO’s 3C model, which encompasses confidence, complacency, and convenience, this review seeks to pinpoint the main factors and suggest focused strategies to address them. **Methods**: Following the PRISMA guidelines, we conducted a systematic search in PubMed, Medline, Web of Science, Scopus, Embase, and Global Health. Eligible studies were cross-sectional, peer-reviewed, and examined COVID-19 vaccine hesitancy among college students. Covidence was used for screening, and data were synthesized narratively using the 3C model. **Results**: Sixty-seven studies (n = 88,345 participants) from 25 countries were included in this study. Confidence factors were the most influential, with fear of side effects (87.18%) and doubts about efficacy (72.4%) as primary concerns. Complacency factors included a low perceived risk of infection (34.9%) and a preference for alternative preventive measures (52.3%). Convenience barriers involved financial costs (58.1%) and difficulty accessing vaccination centers (40.3%). Subgroup analyses revealed variations by academic discipline and geographic region, with medical students showing hesitancy despite their health knowledge. **Conclusions**: COVID-19 vaccine hesitancy among college students is primarily driven by safety concerns, misinformation, and accessibility barriers. Addressing hesitancy requires transparent risk communication, policy-driven accessibility improvements, and tailored educational interventions. These findings can inform strategies to enhance vaccine uptake among young adults and contribute to broader efforts in pandemic preparedness.

## 1. Introduction

Vaccine development is one of the most significant achievements in medical history, saving countless lives [1]. The World Health Organization (WHO) estimates that vaccines prevent 350,000 to 500,000 deaths annually from diseases like diphtheria, tetanus, pertussis, influenza, and measles [2]. The effectiveness of vaccines is indisputable, yet the implementation of vaccines faces numerous challenges. In 2019, vaccine hesitancy was identified as one of the top 10 major global health challenges by the WHO [3]. Vaccine hesitancy encapsulates the reluctance or outright refusal to undergo vaccination, notwithstanding the accessibility of vaccination services. Despite widespread access to vaccines, reluctance to vaccinate continues to hinder optimal coverage in certain populations [4]. Side effects have always been one of the key drivers of this hesitancy, as concerns over adverse reactions may lead to selective acceptance, delayed vaccination, or outright refusal [5].

Vaccine hesitancy among the younger generation, particularly college students, is a significant concern, as these individuals are critical stakeholders in shaping public health outcomes and hold the capacity to advance initiatives aimed at health equity [6]. Their involvement in public health is not just passive; they are often at the forefront of innovative health interventions and advocacy. Projects such as the Fiji Village Project and Canadian frameworks integrating peace and health demonstrate the substantial impact that student-led efforts can have on public health at both the local and international levels [7,8]. With adequate resources and mentorship, students significantly contribute to public health efforts. This positions them not simply as important contributors, but also as potential leaders in the ongoing effort to improve public health infrastructure and outcomes, particularly in tackling pressing issues like vaccine hesitancy and the prevention of infectious diseases. Furthermore, incorporating college students in vaccination programs enhances their public health awareness, preparing them for future challenges. With better knowledge of vaccine hesitancy, students, as future drivers of the public health field, could effectively promote the prevention of new pandemics in the future.

Despite their critical role in public health, little research has focused on vaccine hesitancy among college students. In this study, we will apply the 3C model, endorsed by the WHO for its simplicity and effectiveness in understanding vaccine hesitancy [9]. Compared to broader models, such as the SAGE WG model [10], the 3C model emphasizes three actionable factors: complacency (believing vaccines are unnecessary), convenience (vaccine access and availability), and confidence (trust in vaccine safety and healthcare) [11,12]. This focused approach facilitates the design of targeted interventions to address hesitancy, especially among younger populations, like college students. This systematic review aims to explore vaccine hesitancy in college students in different countries, particularly in the context of COVID-19, using the 3C model to analyze key contributing factors. These insights will help public health officials and policymakers to design more targeted interventions to improve vaccine uptake in younger populations, based on our stratified results by countries and regions. Furthermore, the research could serve as a foundation for future studies on vaccine hesitancy in other contexts involving vaccine-preventable infectious diseases, some of which may similarly encounter public hesitancy to vaccination. These insights can help to foster broader strategies to strengthen vaccine uptake and control infectious disease outbreaks.

## 2. Methods

### 2.1. Study Design

This systematic review adhered to the Preferred Reporting Items for Systematic Reviews and Meta-Analyses (PRISMA) guidelines and utilized the Synthesis Without Meta-analysis (SWiM) framework to ensure rigorous reporting standards (Figure 1). This review aimed to evaluate factors influencing COVID-19 vaccine hesitancy among college students using the 3C model—complacency, confidence, and convenience. The complete PRISMA checklist can be found in Appendix A.

To maintain a focus on descriptive evidence, only cross-sectional studies that examined vaccine hesitancy in college student populations were included. Cross-sectional studies were prioritized because they could capture a comprehensive snapshot of vaccine hesitancy at a given time.

The inclusion criteria were as follows:

1.Peer-reviewed studies published in English.2.Studies involving adult college student populations.3.Cross-sectional studies examining vaccine hesitancy.4.Studies incorporating the 3C model or reporting factors relevant to complacency, confidence, or convenience.

The exclusion criteria were as follows:

1.Conference abstracts, unpublished manuscripts, or non-English studies.2.Studies on non-college populations or general vaccine acceptance without hesitancy data.3.Intervention-based studies, such as randomized controlled trials (RCTs).

### 2.2. Searching Strategy

Our systematic search strategy was developed by an interdisciplinary team led by Dr. Kam, a public health expert with over 10 years of experience in conducting systematic reviews and evidence synthesis. The team also included two trained research assistants, MZ and WYMH, both holding a Bachelor of Science in Public Health and formally trained in systematic review methodology, including database searching, data extraction, and quality appraisal. In addition, a group of dedicated research volunteers (Z.W., S.L., and X.Z.) contributed to various stages of the review process, including citation screening and data organization, under the guidance of senior team members. Each team member brought complementary skills, ensuring that the literature search was both comprehensive and unbiased. The search strategy was formulated and refined through close collaboration among all team members, with Dr. Kam providing final oversight to ensure methodological rigor. Screening of titles and abstracts, as well as full-text reviews, was conducted independently by the trained research assistants, with any disagreements resolved through team discussion.

To identify eligible articles, we conducted a comprehensive search across PubMed, Medline, Web of Science, Scopus, Embase, and Global Health for English-language, peer-reviewed publications. We used the following combination of keywords: “Vaccine hesitancy” OR “Vaccine acceptance” OR “Vaccine refusal” OR “Vaccine reluctance” AND “COVID-19” AND “Confidence” AND “Complacency” AND “Convenience” OR “Social Factors”. The time period of the literature search was from January 1990 to March 2024. The retrieved articles were exported to Covidence for duplicate removal and systematic screening.

The screening process consisted of three steps. First, we imported 1026 articles from PubMed, 216 articles from Medline, 2968 articles from Web of Science, 760 articles from Scopus, and 1334 articles from Global Health, resulting in a total of 6304 articles. Using Covidence, we performed the initial title and abstract screening, eliminating duplicate entries, articles lacking full text, and non-English studies. During this round, 2713 duplicated articles were identified and removed by Covidence, and 3591 studies were included in the initial screening process. Abstracts were manually reviewed based on predefined inclusion criteria. After this round, 137 articles were selected, and 3454 articles were excluded as irrelevant.

In the second step, we conducted a full-text review of the remaining articles, applying the same inclusion criteria. This step led to the exclusion of 70 articles, leaving 67 articles for data extraction. Two reviewers independently screened the full texts, while a third reviewer (W.M., M.Z., or Z.W.) resolved any disagreements that arose during the process. The final 67 articles were included in the data extraction phase, where they were systematically analyzed according to the study framework.

### 2.3. Data Extraction

The data extraction was conducted using Microsoft Excel. Relevant information from the studies included was collected where applicable. Two reviewers independently extracted data, with discrepancies resolved by a third reviewer. Consensus was achieved through discussion. The extracted data were then used to perform a narrative synthesis of the results (Appendix A).

For all included studies, we gathered basic demographic information, including the article’s introduction (author, year, title, journal, study objective, study type, and setting, as well as the data collection period). We also collected details on the study population (e.g., type of college students, male-to-female ratio, ethnicity, and health status). Additionally, information on vaccination choices, such as vaccine hesitancy, influencing factors, and vaccine doses, was extracted.

To better analyze factors contributing to vaccine hesitancy, we applied the 3C model, focusing on complacency, confidence, and convenience. Complacency was divided into two dimensions: individual complacency, reflecting participants’ perceptions of COVID-19 as a personal risk or serious issue, and community complacency, referring to collective perceptions within a group. Vaccine confidence was analyzed in terms of two key dimensions: effectiveness and side effects. Effectiveness pertains to the vaccine’s ability to prevent illness, and is influenced by clinical trial results, public health messaging, and personal experiences. Concerns about side effects, meanwhile, focus on potential long-term health risks and complications, which are especially relevant for individuals with high health awareness and literacy. These individuals often weigh the risks of adverse reactions against the perceived benefits, particularly for new vaccines with limited long-term data. Meanwhile, we categorized sources of vaccine confidence into two critical dimensions—public information and official authorities. Public information sources, which include local communities, social media, and other low-threshold platforms, are characterized by their accessibility and widespread reach. These platforms allow for the rapid dissemination of information, often reflecting the sentiments and perceptions of the general population. However, the credibility of such sources can vary widely, and the information shared may not always be verified or accurate. On the other hand, official authorities, encompassing government bodies, healthcare professionals, and other established institutions, represent a more structured and authoritative source of information. These entities are typically regarded as more reliable due to their expertise and responsibility in public health matters. The distinction between these two categories is crucial because it reflects the differing levels of trust and influence that each source may have on public perceptions and behavior regarding vaccination.

In analyzing vaccine convenience, the categorization into three dimensions—accessibility, availability, and support services—was guided by the need to comprehensively assess the factors that influence an individual’s decision to receive a vaccine. Accessibility refers to the physical proximity and ease of reaching vaccination centers, which is critical for ensuring that people can receive a vaccination without facing significant logistical barriers. Availability focuses on the consistent supply of vaccines, ensuring that there is sufficient stock to always meet demand. Lastly, support services encompass the operational aspects that facilitate vaccination, such as extended operating hours, the presence of healthcare professionals to address concerns, and additional assistance like transportation or information dissemination. These categories were selected based on their relevance in previous research that identifies logistical and operational challenges as major determinants of vaccine uptake [13].

The extracted data were systematically analyzed using the 3C framework to categorize factors influencing vaccine hesitancy. Complacency was assessed through individual and community perceptions of risk and the perceived necessity of vaccination. Confidence encompassed trust in vaccine safety, efficacy, and the reliability of information sources. Convenience focused on logistical barriers, including accessibility, affordability, and support services. Each factor’s significance was quantified by aggregating percentages reported in multiple studies and their corresponding participant counts.

The data underwent several preprocessing steps to ensure accuracy and consistency. Text fields were cleaned to standardize variable names and remove inconsistencies. Percentages embedded in text strings were extracted using automated parsing techniques. Duplicate or redundant information across studies was identified and resolved. Missing data (70 studies) were managed by excluding incomplete rows where necessary, focusing only on factors with sufficient representation across studies. Factors were then categorized and aligned with the 3C dimensions. Finally, the cleaned and structured data were visualized in the form of charts and tables, providing an intuitive summary of key findings. Python 3.11 was utilized throughout this process to automate data cleaning, extraction, and alignment, ensuring precision and reproducibility in the analysis.

### 2.4. WHO 3C Model

The 3C model of vaccine hesitancy considers complacency, confidence, and convenience to be contributing factors, defined as follows: (1) complacency occurs when perceived risks of vaccine-preventable diseases are low and vaccination is not deemed a necessary preventive action; (2) confidence consists of trust in the effectiveness and safety of vaccines and the system that delivers them, including the reliability and competence of the health services and health professionals, as well as the motivations of the policymakers who decide on the needed vaccines; (3) convenience is defined as the extent to which physical availability, affordability and willingness to pay, geographical accessibility, the ability to understand (language and health literacy), and the appeal of immunization services affect uptake. The detailed 3C model of the application context can be seen below in Figure 2.

This conceptual framework is relevant for investigating vaccine hesitancy among college students in the post-COVID-19 pandemic era, as it systematically connects modifying factors, psychological drivers, and targeted interventions to understand and address this issue.

### 2.5. Risk of Bias Assessment

We analyzed several domains (sample size—D1, response rate—D2, confounding factors adjustment—D3, participant selection—D4, and risk of bias in the selection of the reported result—D5). The overall risk of bias for each study was categorized as low, moderate, or high. The summary plot and traffic lighting can be found in the Appendix A (Appendix A).

For domain 1, studies received a “low” risk of bias grade if more than 1000 participants were sampled, a “some concerns” grade if the sample size was between 500 and 1000, and a “high” concern grade if the sample size was under 500. For domain 2, studies received a “low” risk of bias grade if the response rate was above 40%, a “some concerns” grade if the response rate was between 20 and 40%, and a “high” risk grade if the response rates were lower than 20%. If response rates were not published in the study, they received an “unclear” grade. For domain 3, studies received “low” if they adjusted for key confounding factors, “some concerns” if the confounding factors were mentioned but not adequately adjusted for, and “high” if the confounding factors were not identified or adjusted for. For domain 4, studies received “low risk of bias” if participants were randomly selected or the selection method was well-justified and unbiased, “some concerns” if the selection method was not fully random but included steps to minimize bias, or “high risk of bias” if the selection method was likely to introduce bias, like convenience sampling. For domain 5, studies received “low” if they are reported following a pre-determined analysis plan, “some concerns” if some of the selection of the results reported in the study was based on desirability, and “high” if a significant amount of selection of the reported results was based on desirability.

## 3. Results

### 3.1. Characteristics of the Study Population

Figure 3 and Table 1 present an overview of the demographic characteristics and vaccination intentions of the study population, comprising 88, 345 participants. Regarding gender distribution, 37.20% (32,878 individuals) identified as male and 62.80% (55,467 individuals) as female. The male-to-female ratio was approximately 1:1.69, indicating a predominance of female participants.

The academic composition of the population included 73.66% (64,705 individuals) general college students, 16.94% (14,884 individuals) medical students, 4.52% (3975 individuals) healthcare students, and 4.87% (4276 individuals) nursing students. Concerning vaccination intentions, 61.13% (53,694 individuals) expressed willingness to receive the COVID-19 vaccine, 24.24% (21,291 individuals) indicated unwillingness, and 14.63% (12,855 individuals) did not specify their intention.

### 3.2. Country-Wise Distribution of Sample Size

The distribution of the sample size by country is detailed in Table 2. The largest sample sizes were from China (n = 35,812), the United States (n = 10,437), and Germany (n = 7670). Other notable sample sizes included Bangladesh (n = 2909), Ethiopia (n = 2033), Egypt (n = 3204), and Vietnam (n = 2945). The total sample size from all countries was 88,345.

The ranking of influencing factors according to the number of studies mentioned in the total publications (excluding articles without explicit value for the influencing factors) is detailed in Table 3. The most mentioned influencing factors were fear of side effects (n = 34), doubts about vaccine efficacy (n = 20), and preferring to wait for more research due to the quick development of COVID-19 vaccines (n = 16). Safety concerns (n = 12), insufficient information about the vaccine (n = 4), lack of trust in the vaccine (n = 4), distrust in healthcare providers and public health authorities (n = 4), the influence of negative information on social media (n = 4), a preference for other preventive measures (n = 4), and financial cost if the vaccine was not free (n = 4) were also some remarkable factors that hindered people from receiving the vaccination. The ranking based on the number of studies sheds light on participants’ most common concerns about COVID-19 vaccines and reveals the concerns of the minority, which are also worthy of study.

The mean value and standard deviation were other criteria used to evaluate the significance of factors. For example, 50.25% of the studied population thought that fear of side effects was a driving factor of their vaccine hesitancy. According to the table, the most influential factors were financial cost (58.13%, SD = 19.04), insufficient information about the vaccine (55.68%, SD = 19.94), preferring to wait for more research due to the quick development of COVID-19 vaccines (52.28%, SD = 23.04), fear of side effects (50.25%, SD = 20.33), and a lack of trust in the vaccine (49.98%, SD = 19.42). Compared to the number of studies, the rank according to the mean value of percentages was slightly different. The standard deviation should also be considered when evaluating confidence in the mean listed above.

### 3.3. Subgroup Analysis of Vaccine Influencing Factors Among College Students by Academic Discipline

Utilizing the data collected in Table 2, the study conducted a subgroup analysis of vaccine influencing factors among college students across different academic majors, including general college students, medical students, nursing students, and students of other health professions (Figure 4A). The results reveal the most cited influencing factors and some uncommon factors that influenced the decision on whether to take the COVID-19 vaccine among college students from different academic backgrounds, which is reflected in the number of studies reported and shown in the form of heat maps in Figure 4A.

Among students from different fields of study, fear of side effects and doubts about vaccine efficacy were universally prominent, indicating widespread concern about potential adverse effects (Figure 4A). A preference to wait for more research due to the quick development of COVID-19 vaccines and safety concerns were also significant among students. Medical students were equally concerned about side effects and vaccine efficacy. Compared to general college students, medical and nursing students showed stronger concern about the efficacy and safety of the vaccine, while students of other health professions had a stronger willingness to wait for more research and clinical trials on the vaccine. Despite having more medical knowledge, these students were still uncertain about the vaccines’ effectiveness and safety. Moreover, distrust in healthcare providers and public health authorities, insufficient information about the vaccine, personal freedom, the influence of social media (negative information), and a perceived low risk of infection were also identified as significant driving factors, regardless of fields of study. Tailored communication and education strategies are essential to address the specific concerns of each group in order to improve vaccine uptake.

### 3.4. Subgroup Analysis of Vaccine Influencing Factors Among College Students by Regions of Origin

In Figure 4B, the heat map illustrates the percentage of factors influencing COVID-19 vaccine hesitancy among college students from different geographical regions, which is categorized under the 3C framework of confidence, complacency, and convenience. Our study conducted a subgroup analysis of college students across four regions: Asia, Africa, Europe, and America (USA). The results highlight the most cited influencing factors that influence vaccine uptake among college students in each region, as determined by the number of studies that reported these factors, as shown in heat map Figure 4B.

In Asia, according to Figure 4B, side effects were highlighted in 14 studies, making this factor the most prominent barrier to the COVID-19 vaccine, followed by doubts about vaccine efficacy (7 studies) and safety concerns (6 studies). Similarly, in Africa, side effects were the primary concern, mentioned in six studies, with doubts about efficacy and the need for further research also noted. In Europe, safety concerns were the most frequently reported factor (six studies), indicating a regional emphasis on vaccine safety. Other notable concerns included distrust in the vaccine, the influence of negative information from social media, and a perception of the vaccine as unnecessary. In the USA, side effects were again the leading concern, cited in four studies, with additional issues such as the perceived rapidity of vaccine development, personal freedom, and distrust in healthcare providers appearing less frequently.

Side effects were the most frequently cited concern across all regions, particularly in Asia (14 studies) and Africa (6 studies). Safety concerns, doubts about vaccine efficacy, and the perceived speed of vaccine development also appeared as significant barriers in multiple regions.

### 3.5. Analysis Using the 3C Model 

This study systematically analyzed vaccine hesitancy factors using the 3C framework—complacency, confidence, and convenience—and employed dual thresholds (≥50% for major factors and 25–49.9% for minor factors) to prioritize findings. The ≥50% threshold was based on the principle of majority influence, identifying dominant factors with widespread impact [14,15]. Meanwhile, the 25–49.9% threshold captured context-specific barriers aligned with minority influence theories, where smaller but persistent factors can significantly affect subgroups [14]. This dual approach reflects an epidemiological practice of categorizing determinants by prevalence in order to prioritize interventions while ensuring inclusivity [16]. The analysis provides a comprehensive framework for understanding and addressing vaccine hesitancy by integrating these theoretical principles.

By employing dual thresholds, we identified 44 major factors and 12 minor factors influencing vaccine uptake. Confidence-related factors were the most prominent, with 87.18% of respondents expressing fear of side effects [17,18,19,20,21] and 72.4% citing doubts about vaccine safety and efficacy [17,18,19,20,22,23,24]. Complacency was also evident, as 88.8% of participants expressed concerns about infecting others [18,20,23,24,25,26], yet 6.3% believed that “nothing bad would happen” if they contracted COVID-19. This reflects psychological distance, where individuals acknowledge community risks, but underestimate personal vulnerability. Convenience factors, such as difficulties in accessing vaccination centers (40.3%) [21,24,27] and financial barriers (28%) [18,22,27], highlighted the logistical challenges faced by some groups. The major factors contributing to vaccine hesitancy among college students are summarized in the accompanying Figure 5. These visualizations categorize the key drivers under the 3C framework—confidence, complacency, and convenience—highlighting their respective prevalence and significance in influencing vaccine hesitancy.

Complacency reflects the underestimation of risks associated with vaccine-preventable diseases. Across nine studies, 88.8% of participants reported concerns about infecting others [17,18,20,21,23,24,25,26,27], emphasizing an understanding of community risk. However, individual complacency was more pronounced, with 6.3% of respondents believing “nothing bad would happen” if they contracted COVID-19 [24,25,26], and 5.4% perceiving vaccines as unnecessary [24,26].

Lack of trust in vaccine safety and efficacy significantly influenced hesitancy. Fear of side effects was the most frequently cited concern, averaging 87.18% across 10 studies, with reported ranges from 70% to 90% [17,18,19,20,21,22,23,24,26,27]. Similarly, doubts about vaccine safety and efficacy were reported by 72.4% of participants in eight studies [17,18,19,20,22,23,24,25] reflecting skepticism surrounding the rapid development of COVID-19 vaccines. Additionally, insufficient information about vaccines was noted in 74.17% of participants across six studies [18,19,20,22,23,24]. Social media influence was also a significant contributor, with 30% of respondents across five studies citing negative information as a driver of hesitancy [20,22,23,25,28].

## 4. Discussion

Under the 3C model, complacency refers to individuals underestimating their personal risk of contracting COVID-19, and the lower perceived personal risk results in reduced adherence to preventive measures like vaccination and social distancing, potentially leading to lower overall vaccine uptake. This study found that concerns about community transmission were low, aligning with the concept of psychological distance, where individuals feel emotionally or cognitively detached from the societal consequences of their actions [29]. This psychological distance reduces personal accountability for collective health outcomes [30]. The data reveal that many participants in this study expressed moderate concern about the personal impact of the virus, demonstrating a significant level of complacency. This finding aligns with those of previous studies, particularly in younger populations, who often perceive themselves as less vulnerable to health risks [31]. Addressing complacency requires interventions that connect individual actions to broader community outcomes. Vaccination-promoting educational campaigns should integrate real-world narratives (e.g., emphasize examples of specific disease complications) and risk communication strategies to highlight the consequences of vaccine-preventable diseases [32], and highlight individuals’ roles, accountability, and significance in protecting vulnerable groups. Universities and public health organizations could collaborate to develop mandatory health literacy programs, leveraging behavioral science insights to encourage vaccine uptake. Mobile health applications and gamified awareness campaigns could also be applied to spread awareness of the risks associated with non-vaccination [29].

Vaccine confidence is a major driver of hesitancy, especially in relation to fears of side effects and doubts about vaccine efficacy. This finding mirrors previous research, which frequently cites fear of adverse effects, particularly regarding the rapid development of COVID-19 vaccines, as a key barrier to vaccination [32]. Media coverage of rare adverse events, such as blood clotting associated with the AstraZeneca vaccine, likely exacerbated these fears, fueling public apprehension [33]. Research has shown that transparent communication about vaccine side effects, coupled with evidence of their rarity and mildness, can significantly mitigate vaccine hesitancy [4,34]. Thus, focused public health strategies are essential [35]. To counteract misinformation and build confidence, universities could establish trusted information hubs staffed by healthcare professionals and student ambassadors to provide peer-reviewed resources on vaccine safety. Additionally, social media campaigns that feature testimonials from students and medical experts could be deployed to counteract vaccine misinformation. In terms of convenience, the study found that proximity to vaccination centers was the most frequently cited influencing factor on vaccine uptake, a finding consistent with previous studies [36,37]. Policy measures ought to expand mobile vaccination clinics at college campuses, lengthen their hours of service, and offer financial motivations like subsidies for transport. These could include integrating vaccine costs into health insurance or providing modest monetary rewards for vaccinated students. These findings suggest that more research is needed to dispel people’s misgivings about vaccines. At the same time, the authorities and researchers should provide the public with timely updates on vaccine research and keep the information transparent. Universities should also work with policymakers to implement digital vaccine appointment systems that streamline the registration process and reduce any potential logistical barriers. Negative information on social media and financial costs exhibit moderate-to-high levels in some regions, suggesting that socioeconomic conditions and online discourse can both play a role in vaccine decision-making. These findings suggest that addressing concerns related to vaccine safety and efficacy and providing transparent and accessible information about the vaccine development process may be key strategies in improving vaccine uptake among college students globally.

Interestingly, the study shows that even among medical students, who generally possess more advanced health knowledge, concerns about vaccine efficacy and side effects persist. The differences between medical, nursing, and other students highlight a spectrum of vaccine hesitancy influenced by both scientific understanding and broader social factors. Even students receiving health-related education, such as those in medical and nursing programs, exhibited hesitancy, suggesting that formal health-related education alone is insufficient to address underlying concerns. General college students, meanwhile, may be more susceptible to misinformation and public distrust of authorities. Future interventions should consider these nuances, tailoring communication strategies to address both factual concerns (e.g., long-term safety) and broader societal influences (e.g., personal freedom, misinformation). This finding challenges the assumption that increased medical understanding naturally leads to higher vaccine confidence. Medical students, despite their professional background, may share similar hesitations to the general population when faced with uncertainty about long-term vaccine safety [38]. The high representation of general college students suggests that the results are broadly applicable to the general student population. Additionally, the notable proportions of medical students and students of healthcare-related subjects emphasize the relevance of the findings to future healthcare professionals. Our findings emphasize the need for targeted communication strategies that address not only factual knowledge, but also emotional and psychological factors influencing vaccine decisions. The study population’s greater dominance of females may reflect greater interest in or concern about COVID-19 vaccination among women.

These findings indicate that scientific knowledge alone is insufficient to address confidence deficits. Building vaccine confidence requires transparent and accessible communication, highlighting real-world safety and efficacy data. Leveraging trusted healthcare professionals, educators, and community leaders to counter misinformation and provide credible information is essential for mitigating concerns. This gap in current studies points to a need for more comprehensive data collection on logistical barriers to vaccination in future research [34].

Although the World Health Organization has declared the end of the pandemic phase of COVID-19 [39], vaccine hesitancy remains a pressing public health issue with implications beyond the pandemic itself. While this study focused on COVID-19 vaccine hesitancy among college students, the findings of our research can also be broadly applied to other diseases, such as Disease X, which represents an unknown pathogen with the potential to cause future pandemics, as defined by the WHO [40,41,42]. By continuously focusing on vaccine hesitancy in the post-pandemic era, this study contributes to strengthening long-term public health resilience and promoting proactive immunization strategies.

Future studies should extend beyond COVID-19, exploring hesitancy toward routine vaccines (e.g., HPV, influenza) and preparedness for emerging pathogens and emerging infectious threats (e.g., “Disease X”). Comparative studies examining whether the same psychological and logistical barriers apply across different vaccines would provide valuable insights into persistent hesitancy trends. Additionally, further research is needed to understand how cultural factors influence vaccine acceptance among college students in different regions, as well as the long-term effects of misinformation on vaccine decision-making. Longitudinal studies tracking vaccine attitudes over time could help to assess whether hesitancy decreases as more students enter the workforce or health professions. Future research should also explore the effectiveness of targeted interventions, such as peer-led education programs or social media fact-checking campaigns, in increasing vaccine confidence.

## 5. Limitations

The current study is subject to several limitations that merit further investigation. Our focus on COVID-19 vaccine hesitancy may limit the generalizability of our findings to other vaccine types. The primary factor of confidence in vaccine safety is particularly influenced by the rapid development and emergency deployment of COVID-19 vaccines. This context may differ from that of routinely administered vaccines, especially among well-educated college students.

While we presented the country-wise distribution using absolute participant numbers, we did not adjust the sample sizes relative to the total national populations. Such adjustments could provide additional insights in studies seeking generalizable population-level inferences. However, given our targeted subgroup of college students, absolute participant numbers provide the most direct and meaningful measure relative to our research objectives. Future research aiming at broader population-level generalization may benefit from incorporating population-adjusted measures to enhance interpretability across different countries. Third, there was notable under-representation of studies from Latin America and countries predominantly speaking Romance languages (e.g., Spanish, Portuguese, and Italian). With France being the only country with significant representation among these countries, this regional gap may have introduced bias and may reduce the global applicability of our findings. Vaccine hesitancy may manifest differently due to distinct cultural, economic, and political influences present in these under-represented areas.

Female students accounted for 63.15% of the sample, which may reflect greater willingness among women to participate in health-related surveys. This skew could also introduce bias if gender differences significantly influence vaccine attitudes.

A summary of the risk of bias across studies revealed that approximately 60% of included studies were assessed as having a low risk of bias, 25% had a moderate risk, and 15% had a high risk of bias. Most concerns stemmed from small sample sizes and non-random sampling methods. These variations may impact the generalizability of the findings.

Additionally, logistical factors, such as vaccine supply, operational hours, and support services, were less frequently mentioned [37]. This under-reporting could suggest that, while these elements are critical, they may be overshadowed by the more immediate and tangible concerns of accessibility, such as proximity to vaccination centers. This tendency to focus on direct access might result from the relatively straightforward nature of measuring the distance to vaccination sites compared to the more complex evaluation of operational hours and service quality.

Furthermore, these factors might be perceived as secondary considerations, particularly in regions where proximity itself remains a major barrier. As a result, logistical barriers like clinic hours and available support services are often underexplored in research, indicating a significant gap in the literature. Addressing this gap through more comprehensive data collection is essential, as extending clinic hours and enhancing support services like transportation could play a critical role in increasing vaccine uptake, particularly in underserved populations [13]. The lessons from COVID-19 vaccine hesitancy can inform strategies to improve uptake for routine immunizations among younger populations, including HPV, influenza, and meningococcal vaccines. Moving forward, longitudinal studies and real-time intervention trials are needed to assess whether targeted communication and accessibility improvements can sustainably reduce vaccine hesitancy in both pandemic and non-pandemic settings.

Finally, although we conducted a thorough search of six major databases, we did try to reach out to corresponding authors of inaccessible full-text articles, though did not receive a response from many of them. As such, it is possible that some eligible studies were missed due to full-text unavailability, which may have introduced selection bias and affected the comprehensiveness of the review.

Future studies should investigate the nuanced impact of these logistical elements to provide a more holistic understanding of the barriers to vaccination. Future research should also actively seek broader geographic and cultural representation, including Latin American and other Romance language-speaking countries, to better capture the diverse factors influencing vaccine hesitancy globally. It should also investigate the nuanced impact of these logistical elements to provide a more holistic understanding of the barriers to vaccination.

## 6. Conclusions

This study applied the 3C model to understand COVID-19 vaccine hesitancy, focusing on college students. Currently, vaccine hesitancy remains relatively high, and there are several reasons for this hesitancy. The most common reasons include fear of side effects, doubts about vaccine efficacy, an overly rapid development process, and safety concerns. The recurring patterns of vaccine hesitancy observed in past outbreaks (e.g., H1N1, MMR, influenza) suggest that similar barriers will persist in future pandemics. This phenomenon implies that vaccine hesitancy is still a complicated and rapidly changing global issue, slowing down the process of controlling various vaccine-preventable diseases, which are among the top ten threats to global health. This complex problem requires multiple interventions to meet different specific needs from the aspects of convenience, complacency, and confidence individually. This could include integrating community healthcare workers to optimize the management of the disease, as has been shown to be effective in previous research [43]. Clear and engaging communication with the public is crucial to dispel myths, alleviate fears, respond to queries, and promote the acceptance of vaccination. Moreover, vaccine hesitancy is not solely a problem that happens in a selected area or a special period. This study on COVID-19 vaccine hesitancy can serve as a foundation for future studies on vaccine hesitancy, and can also be applied to other vulnerable populations and diseases in the post-pandemic period [44].

In summary, while this study focuses on COVID-19 vaccines, its findings have broader implications for understanding and combating vaccine hesitancy in future public health emergencies. Addressing vaccine hesitancy should be an ongoing priority, requiring multifaceted strategies that integrate scientific evidence, public trust, and equitable vaccine access to ensure high uptake in future vaccination efforts.

## Figures and Tables

**Figure 1 vaccines-13-00461-f001:**
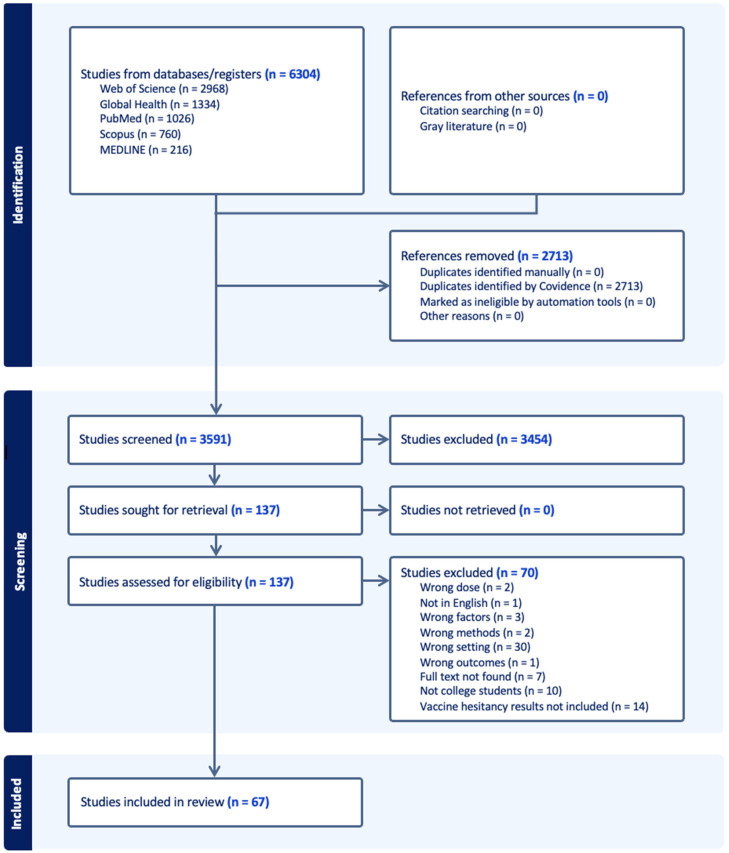
PRISMA flowchart for information search in Web of Science, Global Health, PubMed, Scopus, and MEDLINE. Diagram shows selection of reports included in review.

**Figure 2 vaccines-13-00461-f002:**
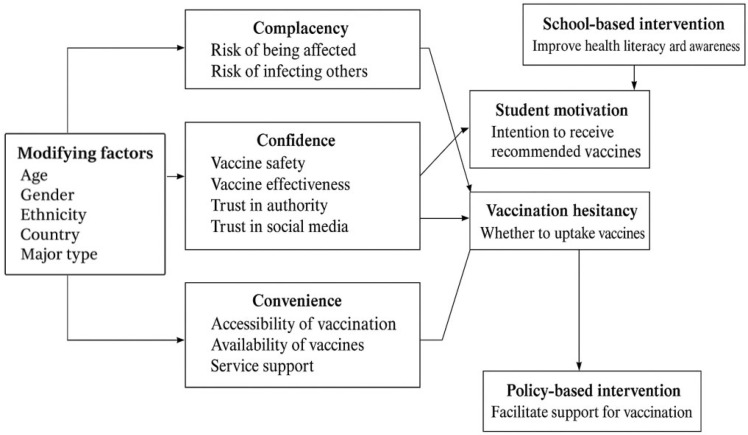
Framework of 3C model.

**Figure 3 vaccines-13-00461-f003:**
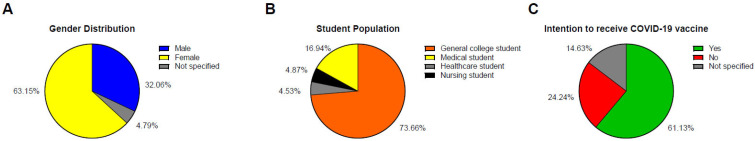
Basic characteristics of the study population and their intention to receive the COVID-19 vaccine. (**A**) This pie chart shows the gender distribution among participants. Participants were categorized into male, female, and not specified. (**B**) This pie chart shows the student population’s constitution based on different academic disciplines. Students were classified according to their declared majors during their undergraduate study. (**C**) This pie chart shows participants’ intention to receive the COVID-19 vaccine. Participants were categorized based on their answers: yes, no, or not specified.

**Figure 4 vaccines-13-00461-f004:**
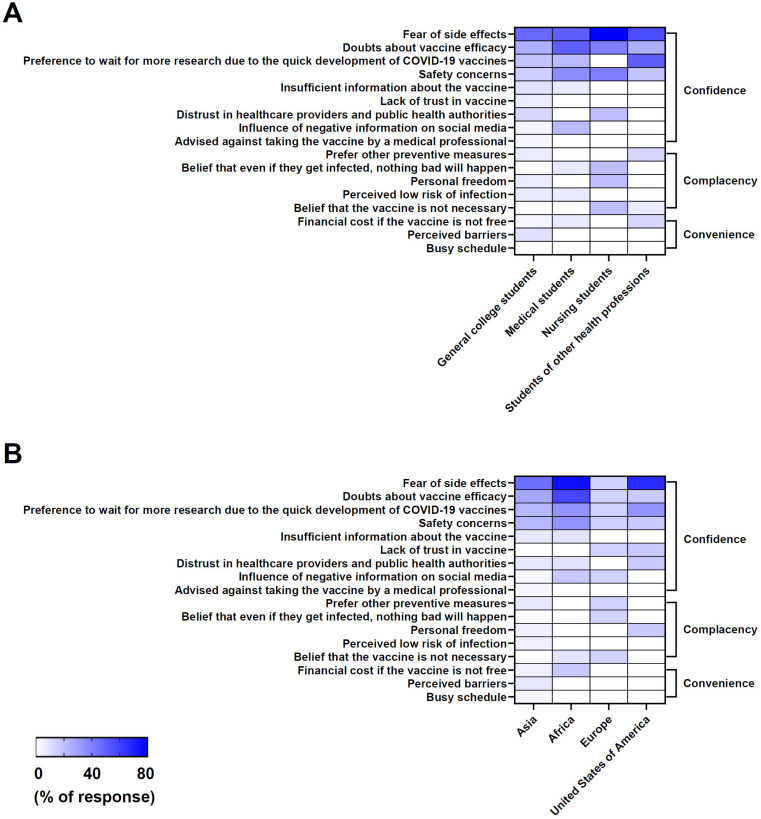
Subgroup analysis of vaccine influencing factors among college students. (**A**) The heat map shows the percentage of factors influencing COVID-19 vaccine hesitancy among college students from different academic disciplines. Students were classified according to their declared majors during their undergraduate study. (**B**) The heat map shows the percentage of factors influencing COVID-19 vaccine hesitancy among college students from different geographic locations. Students were categorized by their geographic regions of origin. The vaccine influencing factors were organized by hierarchical clustering based on the 3C framework—confidence, complacency, and convenience. Data are expressed as the percentage of responses about specific influencing factors (percentage of response = 100 × (number of publications reported with specific influencing factors/total number of publications from specific subgroups)) and scaled between 0% and 80% (with a single gradient coloring range from white to purple) for each influencing factor. The heat maps were prepared with GraphPad Prism 10.3.1 software.

**Figure 5 vaccines-13-00461-f005:**
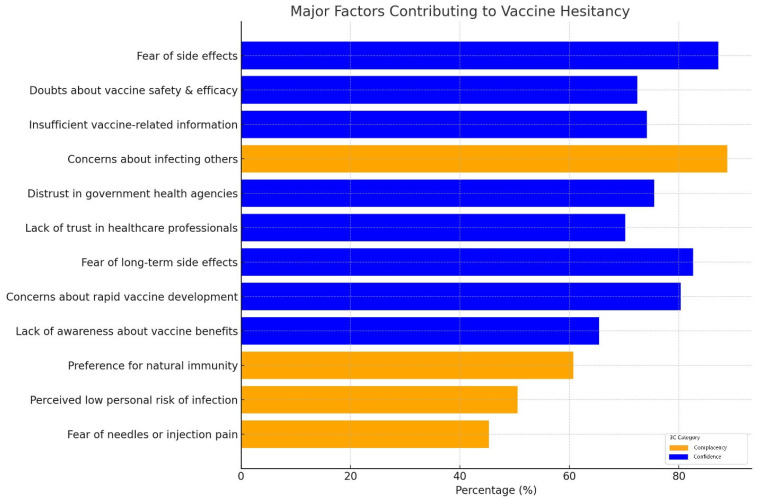
A bar chart illustrating the major factors contributing to COVID-19 vaccine hesitancy among college students, categorized by the 3C framework (confidence, complacency, and convenience). Each bar represents the percentage of respondents, aggregated across multiple studies, who identified a specific factor as a primary reason for hesitancy. The color-coding reflects whether the factor falls under “confidence” (e.g., fear of side effects, doubts about vaccine efficacy) or “complacency” (e.g., perceived low personal risk, preference for natural immunity).

**Table 1 vaccines-13-00461-t001:** Basic characteristics of the study population and their intention to receive the COVID-19 vaccine.

	Distribution	
	Total (n)	Percentage
Gender		
Male	32,878	37.20%
Female	55,467	62.80%
Total sample size	88,345	100%
Male:female ratio	1:1.69	
Student population		
General college student	64,705	73.66%
Medical student	14,884	16.94%
Healthcare student	3975	4.52%
Nursing student	4276	4.87%
Intention to receive COVID-19 vaccine		
Yes	53,694	61.13%
No	21,291	24.24%
Not specified *	12,855	14.63%

Notes: * Not specified means respondents that did not provide a specific yes or no answer, which indicates the proportion of respondents who were hesitant to take the vaccine, but did not completely refuse to be vaccinated.

**Table 2 vaccines-13-00461-t002:** Distribution of total study sample sizes by country.

Country	Total Sample Size	Number of Studies
Australia	1670	2
Bangladesh	2909	4
China	35,812	12
Croatia	752	1
Ethiopia	2033	5
Egypt	3204	2
France	1465	1
Finland	725	1
Germany	7670	2
Greece	2249	1
India	1723	2
Israel	732	2
Japan	1776	1
Lebanon	800	1
Malaysia	1274	1
Morocco	1272	1
Nigeria	440	1
Qatar	364	1
Pakistan	415	1
Poland	872	1
Saudi Arabia	2040	4
Sharjah	669	1
Sudan	217	1
Thailand	409	1
Turkey	1069	1
United Arab Emirates	1476	3
United States	10,437	6
Uganda	600	1
Vietnam	2945	5
Zambia	326	1
Total	88,345	67

**Table 3 vaccines-13-00461-t003:** Ranking of influencing factors on COVID-19 vaccine hesitancy.

Category	Influencing Factor	Number of Studies	Mean Value (SD)
Confidence	Fear of side effects	34	50.25% ± 20.33
Confidence	Doubts about vaccine efficacy	20	35.29% ± 28.30
Confidence	Preference to wait for more research due to the quick development of COVID-19 vaccines	16	52.28% ± 23.04
Confidence	Safety concerns	12	48.33% ± 26.73
Confidence	Insufficient information about the vaccine	4	55.68% ± 19.94
Confidence	Lack of trust in vaccine	4	49.98% ± 19.42
Confidence	Distrust in healthcare providers and public health authorities	4	32.73% ± 25.69
Confidence	Influence of negative information on social media	4	32.30% ± 13.98
Complacency	Preference for other preventive measures	4	52.36% ±21.53
Convenience	Financial cost if the vaccine is not free	4	58.13% ± 19.04
Convenience	Perceived barriers	3	43.00% ± 12.59
Complacency	Belief that even if they are infected, nothing bad will happen	2	27.90% ± 30.55
Complacency	Personal freedom	2	23.80% ± 25.17
Complacency	Perceived low risk of infection	2	34.90% ± 29.27
Complacency	Belief that the vaccine is not necessary	2	8.40% ± 3.00
Confidence	Advised against taking the vaccine by medical professional	1	15.80% ± 0.00
Convenience	Busy schedule	1	39.70% ± 0.00

Abbreviation: SD—standard deviation. Notes: eighteen articles were excluded because they did not provide numerical statistics for influencing factors; total number of studies N = 67. The mean value (%) indicates the average percentage of respondents (across included studies) identifying each factor as influencing their vaccine hesitancy.

## Data Availability

The data that support the findings of this study are available in the Appendix A of this article.

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
