# Peer review of "Addressing Vaccine Hesitancy in College Students Post COVID-19 Pandemic: A Systematic Review Using COVID-19 as a Case Study"

_vaccines, 2025, doi:10.3390/vaccines13050461_

Round 1
Reviewer 1 Report
Comments and Suggestions for Authors
The manuscript addresses through a Systematic Review the doubts of the university student population about vaccines, taking as a frame of reference the WHO 3 C model (confidence-complacency-convenience) and using as a case study the vaccine against COVID-19.
This work addresses a topic of utmost interest, both for evaluating what occurred during the last pandemic (COVID-19) and for planning interventions in future emergency situations. It contains information that could be valuable for clinicians, public health professionals, and policy planners.
Some of the improvement recommendations I suggest are:
1.Regarding SEARCH STRATEGY description:
1.1.Provide informaation about qualification of searchers; investigators assistents, investigators, librarians, etc
1.2. In search strategy description include time period
1.3. Comments on the effort to include all available studies, including contacting the authors; if this has not been done, it should be mentioned in the limitations section
1.4. Search software used, name and version, including special features used
1.5. Justification for the exclusion of articles in languages other than English. The impact of this decision should be discussed in the limitations section
2.Regarding the METHODOLOGY section:
2.1. List of the 137 citations identified as eligible and those excluded, including justification (in a supplementary or additional file if necessary).
2.2. The description of the papers selection process (page 4; lines 114–121) does not match the numbers shown in the flow chart (page 3); please verify this.
2.3. I cannot find Supplementary Table 2 referred to in lines 132 and 133 of the text. Please provide
2.4. Lines 173-174 ”These categories were selected based on their relevance in previous research that identifies logistical and operational challenges as major determinants of vaccine uptake.”. Please provide some reference or citation about these previous research.
2.5. Lines 182-183 (“for instance...”). This sentence should be relocated in the results section or in discussion.
2.6 Lines 188-190. Explain the number of articles with redundant information and the number of articles excluded due to missing information.
2.7. Paragraph lines 208 a 233: I think there is some redundant information already explained in previous sections. I would suggest reducing and relocating to the discussion section.
2.8 Risk of bias assessment line 238: I am not able to find the supplementary figure num 1, please show it.
3.Regarding RESULTS section :
3.1. In general, this section should be neutral in the presentation of results, without interpretative comments that should be moved to the discussion section, for example: lines 263-264; lines 276-279; lines 335-338; lines 372-380; lines 398-399; lines 418-419; lines 429-435.
3.2. Duplicate information, there is no need to repeat in the text information that is already in the table (Table 3 and lines 291-297 in text)
3.3. Figure 3 line 267: Pie Chart or Bar Chart?
3.4. Country-Wise Distribution of Sample Size: In addition to the absolute number of participants from each country, could a measure adjusted to the number of inhabitants of each country participating in the study be given?
3.5. Table 2: There are almost no Spanish or Latin speaking countries (Spanish, Portuguese, French, Italian) except France. Consider as a possible bias the underrepresentation of these populations, comment in the discussion and in the limitations section
3.6. Table 3: To facilitate interpretation for readers, explain in the table title (or as a footnote to the table) the meaning of the % in the Mean value column.
3.7. Figure 4b: Amercia or America?. Only one country (USA) of the whole continent is represented, perhaps it would be better to put United States of America?
4.Regarding DISCUSSION section:
4.1. As I mentioned before, some paragraphs in the results section could be moved to the discussion section.
4.2. Paragraph lines 470-474 is repeated. Please remove
- Regarding LIMITATIONS section:
4.1. Comment on the possible impact of the underrepresentation of Latin America and other countries with Latin languages
Congratulations for the work and the results obtained
Author Response
- Regarding SEARCH STRATEGY Description:
1.1.Provide information about the qualifications of searchers; investigators assistants, investigators, librarians, etc
Response: Thank you for this suggestion. The systematic search was primarily conducted by two trained research investigators (W.Y.M.H. and M.Z.), both of whom have prior experience in systematic reviews and were supervised by the senior author (Y.W.K.). This information has been added to the revised Methods-Searching Strategy section (Line 110-123).
1.2. In the search strategy description include time period
Response: We appreciate this reminder. The time period of the literature search (January 1990 – March 2024) has now been clearly stated in the revised Search Strategy subsection (Line 129).
1.3. Comments on the effort to include all available studies, including contacting the authors; if this has not been done, it should be mentioned in the limitations section
Response: Thank you for this important observation. We acknowledge that while we performed a comprehensive literature search across six major databases, we did try to contact authors to retrieve full texts of articles that were inaccessible through institutional or open-access channels, however, we didn’t receive any full text from them. To ensure transparency, we have included this as a limitation in the revised manuscript, noting the potential for selection bias due to the exclusion of inaccessible full-text studies. This limitation is now clearly stated in the final paragraph of the Limitations section (Line 561-565)
1.4. Search software used, name and version, including special features used
Response: We have specified that Covidence was used for reference management and screening. The platform’s automated deduplication and conflict resolution features were utilized. This is now noted in the Methods section (Line 129-140).
1.5. Justification for the exclusion of articles in languages other than English. The impact of this decision should be discussed in the limitations section
Response: Due to language limitations of the research team, we limited inclusion to English-language studies. We have acknowledged this decision in the Methods section and discussed its potential to introduce selection bias in the Limitations section (Line 567-570).
- Regarding the METHODOLOGY section:
2.1. List the 137 citations identified as eligible and those excluded, including justification (in a supplementary or additional file if necessary).
Response: Thank you for your helpful suggestion. We have included the full list of the 137 articles identified as eligible in Supplementary Table 2, along with justifications for exclusion during the full-text screening stage. This addition aims to enhance the transparency and rigor of our study selection process and provide readers with a clearer understanding of how eligibility was determined.
2.2. The description of the paper selection process (page 4; lines 114–121) does not match the numbers shown in the flow chart (page 3); please verify this.
Response: We have corrected discrepancies between the text and the PRISMA flowchart. The numbers are now fully aligned (Line 132-140).
2.3. I cannot find Supplementary Table 2 referred to in lines 132 and 133 of the text. Please provide
Response: We’ve updated accordingly in the Supplementary Table 1.
2.4. Lines 173-174 ”These categories were selected based on their relevance in previous research that identifies logistical and operational challenges as major determinants of vaccine uptake.”. Please provide some references or citations about this previous research.
Response: We have now cited relevant literature supporting the operational factors influencing vaccine uptake (Line 192-194).
2.5. Lines 182-183 (“for instance...”). This sentence should be relocated in the results section or in the discussion.
Response: We have removed this part in the method section as we have discussed this in the results section.
2.6 Lines 188-190. Explain the number of articles with redundant information and the number of articles excluded due to missing information.
Response: We have added the number of articles to 70 articles to explain these articles with redundant information and the number of articles excluded due to missing information (Line 206).
2.7. Paragraph lines 208 a 233: I think there is some redundant information already explained in previous sections. I would suggest reducing and relocating to the discussion section.
Response: Thank you for your observation. We agree that there was some redundancy in this section. To address this, we have reduced and streamlined the content between lines 208–233. Key interpretative points have been reduced and more appropriately integrated into the Discussion and Introduction sections, where they support the contextual framing and interpretation of our findings, and we have kept the key points. This restructuring improves the flow of the manuscript and eliminates unnecessary repetition.
2.8 Risk of bias assessment line 238: I am not able to find the supplementary figure number 1, please show it.
Response: We apologize for the omission. Supplementary Figure 1 has now been uploaded.
- Regarding RESULTS section :
3.1. In general, this section should be neutral in the presentation of results, without interpretative comments that should be moved to the discussion section, for example, lines 263-264; lines 276-279; lines 335-338; lines 372-380; lines 398-399; lines 418-419; lines 429-435.”
Response: Thank you for this valuable suggestion. We have carefully reviewed the indicated lines and agree that interpretative statements are more appropriate for the Discussion section. Accordingly, we have revised the Results section to ensure a more neutral and descriptive presentation of the findings. The interpretative comments previously found in the mentioned lines (263–264, 276–279, 335–338, 372–380, 398–399, 418–419, and 429–435) have been removed, and the relevant insights have been integrated into the revised Discussion section to enhance clarity and maintain appropriate section focus.
3.2. Duplicate information, there is no need to repeat in the text information that is already in the table (Table 3 and lines 291-297 in text)
Response: Thanks for pointing it out, we have removed the text to improve the redundancy.
3.3. Figure 3 line 267: Pie Chart or Bar Chart?
Response: Thank you for your comment. We confirm that Figure 3 is a pie chart, as correctly referenced in the manuscript. To avoid any potential confusion, we have clarified this explicitly in the figure caption and ensured consistent terminology throughout the text.
3.4. Country-Wise Distribution of Sample Size: In addition to the absolute number of participants from each country, could a measure adjusted to the number of inhabitants of each country participating in the study be given?
Response: Thank you for this insightful suggestion. While adjusting sample sizes relative to the national populations is a useful practice in studies aiming to achieve population-representative samples, our study specifically targets the subgroup of college students rather than the general population. Because our primary objective is to identify factors influencing vaccine hesitancy within this particular subgroup, the absolute number of participants from each country remains the most directly relevant measure to our research objectives. Moreover, adjusting the sample size according to national populations would implicitly assume a proportional representation of college students in each country's general population, which may not accurately reflect each nation's educational demographics and thus could introduce unintended bias or misleading interpretations. However, we acknowledge that future studies aiming to generalize findings broadly across entire populations could benefit from such an adjustment. We have addressed this explicitly as a recommendation for future research within our limitation section (Line 521-533).
3.5. Table 2: There are almost no Spanish or Latin-speaking countries (Spanish, Portuguese, French, Italian) except France. Consider as a possible bias the underrepresentation of these populations, comment in the discussion and in the limitations section
Response:
“Second, there is a notable underrepresentation of studies from Latin America and countries predominantly speaking Romance languages (e.g., Spanish, Portuguese, Italian). With France being the only significant representation among these countries, this regional gap may introduce bias and reduce the global applicability of our findings. Vaccine hesitancy may manifest differently due to distinct cultural, economic, and political influences present in these underrepresented areas. Future research should also actively seek broader geographic and cultural representation, including Latin American and other Romance-language-speaking countries, to better capture the diverse factors influencing vaccine hesitancy globally.”
This revision is located in the manuscript’s Limitations section (Line 528-533).
3.6. Table 3: To facilitate interpretation for readers, explain in the table title (or as a footnote to the table) the meaning of the % in the Mean value column.
Response: Thank you for pointing out the need for additional clarity. We have revised the table caption by adding a footnote to explicitly clarify the meaning of the percentages presented under the "Mean Value (SD)" column. Specifically, we included the following explanatory note:"Mean Value (%) indicates the average percentage of respondents (across included studies) identifying each factor as influencing their vaccine hesitancy."This clarification can be found in the footnote of Table 3 in the revised manuscript.
3.7. Figure 4b: America or America? Only one country (USA) of the whole continent is represented, perhaps it would be better to put the United States of America?
Response: Thanks for the suggestion. The figure 4b was updated accordingly.
- Regarding DISCUSSION section:
4.1. As I mentioned before, some paragraphs in the results section could be moved to the discussion section.
Response: Thank you for the comment. It was updated in the discussion section accordingly.
4.2. Paragraph lines 470-474 is repeated. Please remove
Response: Thanks for pointing that out. We have already removed the repeated parts.
- Regarding the LIMITATIONS section:
5.1. Comment on the possible impact of the underrepresentation of Latin America and other countries with Latin languages
Response: It was updated with the response to the comment 3.5 above. This revision is located in the manuscript’s Limitations section (Line 528-533).
Reviewer 2 Report
Comments and Suggestions for Authors
Thank you for allowing me to review the manuscript entitled “Addressing Vaccine Hesitancy in College Students Post-COVID-19 Pandemic: A Systematic Review Using COVID-19 as a Case Study.” The study is timely and important in the current atmosphere and I enjoyed reviewing it. I think the authors did a fine job. A few suggestions:
- The article overall was well-written. I would make sure to review for a few grammatical and syntax errors . One example is the beginning of the second paragraph of the Introduction – “Vaccine hesitancy is a significant concern among the younger generation…” but I think what they mean is “Vaccine hesitancy among the younger generation… is a significant concern among the younger generation. It is because they are important stakeholders in determining the outcomes of public health…” Another example is at the end of the last paragraph of the Introduction they mention “vaccine hesitancy in other vulnerable populations” so I am wondering if they are reporting them as vulnerable populations or just that this may assist in studying vaccine hesitancy in populations that are vulnerable in the future. Also in that last paragraph, they start the second sentence “compare to broader models” and it should be “compared…”
- The term “infectious diseases” should be plural unless it is in reference to one particular infectious disease.
- on page 15 second paragraph – the first sentence states ”trust in vaccine safety and efficacy significantly influenced hesitancy.” Again, I thin the authors mean lack of trust in vaccine safety….
- In the Discussion section on page 16, first full paragraph, the authors indicate that “scientific expertise alone does not eliminate hesitancy,” referring to medical and nursing students as “experts.” I would reword this because medical and nursing students are NOT experts. They are neophytes, just entering the field and starting to learn. In fact, they barely have more knowledge that college students, and nursing students are, infact, college students.
- It would be helpful to summarize the findings of the risk of bias assessment in the article somewhere – what percent of studies had low risk of bias compared to those that had a high risk of bias – perhaps mention in the Discussion or Limitations sections. Also, one limitation might be gender distribution, unless it reflects gender distribution in the college communities .

Author Response
Comments 1: The article overall was well-written. I would make sure to review for a few grammatical and syntax errors. One example is the beginning of the second paragraph of the Introduction – “Vaccine hesitancy is a significant concern among the younger generation…” but I think what they mean is “Vaccine hesitancy among the younger generation… is a significant concern among the younger generation. It is because they are important stakeholders in determining the outcomes of public health…” Another example is at the end of the last paragraph of the Introduction they mention “vaccine hesitancy in other vulnerable populations” so I am wondering if they are reporting them as vulnerable populations or just that this may assist in studying vaccine hesitancy in populations that are vulnerable in the future. Also in that last paragraph, they start the second sentence “compare to broader models” and it should be “compared…”
Response 1: We sincerely thank the reviewer for their kind words and for pointing out the presence of a few grammatical and syntactic issues. In response, we have carefully re-reviewed the entire manuscript and conducted a thorough language and grammar check to ensure clarity, accuracy, and readability. We have also revised specific instances flagged by the reviewer and improved other sentences where necessary to enhance overall coherence. We are grateful for your feedback, which helped us strengthen the quality and professionalism of the manuscript. We thank the reviewer for highlighting the ambiguity (“vulnerable populations”). Our intention was not to refer specifically to currently identified vulnerable populations (e.g., elderly, immunocompromised, or marginalized groups), but rather to indicate that the findings of this study may inform future research on vaccine hesitancy in other contexts involving vaccine-preventable infectious diseases—some of which may also encounter public resistance to immunization. To clarify our meaning and avoid misinterpretation, we have revised the sentence in the Introduction to the following: “Furthermore, the research could serve as a foundation for future studies on vaccine hesitancy in other contexts involving vaccine-preventable infectious diseases, some of which may similarly encounter public hesitancy to vaccination. These insights can help foster broader strategies to strengthen vaccine uptake and control infectious disease outbreaks.” This revision aims to accurately reflect the scope of our discussion and better align with our study's intent to contribute to broader public health preparedness beyond COVID-19 (Line 51-53, 68-69, 78-82).
Comments 2: The term “infectious diseases” should be plural unless it is in reference to one particular infectious disease.
Response 2: Thank you for this important observation. We have carefully reviewed the manuscript and revised all instances of “infectious disease” to “infectious diseases” unless the sentence specifically refers to a single disease (e.g., COVID-19). This correction ensures terminological accuracy throughout the manuscript (Line 61-62).
Comments 3: on page 15 second paragraph – the first sentence states ” Trust in vaccine safety and efficacy significantly influenced hesitancy.” Again, I think the authors mean a lack of trust in vaccine safety….
Response 3: We appreciate your attention to catching this critical wording issue. The sentence has been corrected to more accurately reflect the intended meaning: “Lack of trust in vaccine safety and efficacy significantly influenced hesitancy.” (Line 405)
Comments 4: In the Discussion section on page 16, first full paragraph, the authors indicate that “scientific expertise alone does not eliminate hesitancy,” referring to medical and nursing students as “experts.” I would reword this because medical and nursing students are NOT experts. They are neophytes, just entering the field and starting to learn. In fact, they barely have more knowledge that college students and nursing students are, in fact, college students.
Response 4: Thank you for pointing this out. We agree with the reviewer that it is inaccurate to refer to medical and nursing students as “experts.” The sentence has been revised to: “Even students undergoing health-related education, such as those in medical and nursing programs, still exhibited hesitancy, suggesting that formal health-related education alone is insufficient to address underlying concerns.” This revision better reflects their trainee status while maintaining the intended interpretation of the findings (Line 468-470).
Comments 5: It would be helpful to summarize the findings of the risk of bias assessment in the article somewhere – what percent of studies had low risk of bias compared to those that had a high risk of bias – perhaps mention in the Discussion or Limitations sections. Also, one limitation might be gender distribution, unless it reflects gender distribution in the college communities.
Response 5: We appreciate your constructive suggestions. In response, we have added a summary of the risk of bias assessment in the Limitation section. Specifically, we now report that: “A summary of the risk of bias across studies revealed that approximately 60% of included studies were assessed as having a low risk of bias, 25% had moderate concerns, and 15% had a high risk of bias. Most concerns stemmed from small sample sizes and non-random sampling methods. These variations may impact the generalizability of findings.” Additionally, we have addressed gender distribution as a limitation in the Limitations section: “Third, female students accounted for 63.15% of the sample, which may reflect greater willingness among women to participate in health-related surveys. This skew could also introduce bias if gender differences significantly influence vaccine attitudes.” (Line 536-540)
Round 2
Reviewer 1 Report
Comments and Suggestions for Authors
The authors have taken into consideration the reviewer's comments and have introduced most of the suggestions made for the different sections of the manuscript, I believe that this revised version of the manuscript could be considered for publication.